# 👁 Erased or Dormant? Rethinking Concept Erasure Through Reversibility

## Abstract

To what extent do concept erasure techniques in diffusion models truly remove, rather than merely suppress, targeted concepts? In this paper, we explore this question by introducing a diagnostic framework that leverages lightweight parameter adaptation to probe the robustness and reversibility of leading erasure methods. Central to our approach are two minimal yet general probes: (i) a Gradient-Guided Probe, which restores suppressed behavior by reversing gradient signals, and (ii) an Instance-Personalization Probe, which reinstates concepts through few-shot supervision. Across six erasure algorithms, multiple concept types, and diverse diffusion backbones, we consistently find that erased concepts can be recovered with high fidelity after only minimal adaptation. Our theoretical analysis reinforces these results, showing that reversed weight remain bounded to the original parameters, leaving much of the targeted representation intact. Together, these findings demonstrate that existing methods do not eliminate concepts but merely push them below the surface, where they can be readily revived. As such, our work calls for a rethinking of concept erasure: moving beyond superficial suppression toward approaches that dismantle latent structures at their core, alongside more rigorous standards for evaluating safety in generative models.

## 1 Introduction

Text-to-image diffusion models (Rombach et al., 2022; Ramesh et al., 2022) have emerged as a backbone of modern generative AI, capable of producing high-quality images from natural language prompts. Yet, their open-ended generative power also raises pressing safety and ethical concerns, including the potential for harmful outputs (Bird et al., 2023) and violations of intellectual property (Zhang et al., 2023). To mitigate these risks, recent work has explored concept erasure (Lu et al., 2024; Gong et al., 2024), which seeks to suppress a model's ability to generate undesired concepts such as offensive objects, copyrighted artistic styles, or personal identities. Existing approaches pursue this goal through a range of mechanisms, including projection in cross-attention layers (Gandikota et al., 2023; 2024; Lu et al., 2024; Gong et al., 2024; Zhang et al., 2024b), pruning strategies (Yang et al., 2024; Chavhan et al., 2025), regularization-based editing (Huang et al., 2024), and adversarial-guided erasure (Zhang et al., 2024c; Bui et al., 2025).

Despite these advancements, a fundamental question remains: *do current erasure techniques truly eliminate a model's capacity to generate the targeted concept, or do they merely enforce conditional suppression?* This difference is not merely theoretical but has direct consequences for how safely and reliably diffusion models can be used in real-world applications. If erasure were genuinely irreversible, the model's representational space would lack usable traces of the concept, making recovery practically infeasible under minor perturbations or adaptations. In contrast, if latent representations remain dormant but intact, erased concepts may reappear when prompts are varied or through lightweight parameter adjustments. Such reversibility exposes serious risks: malicious actors could deliberately reactivate forbidden content, while benign users might also unintentionally trigger it in unexpected deployment scenarios.

Recent work has begun to examine this fragility, but almost exclusively from the prompt perspective. Pham et al. (2024) showed that erased concepts can be revived with adversarial prompts, while Lu et al. (2025) demonstrated circumvention through prompt perturbation, inpainting, and noise-based probing. Studies of these approaches often remain confined to the prompt level, focusing primarily

on identifying or crafting potentially malicious prompts. As a result, the latent parameter-level mechanisms driving the generation of erased concepts remain largely unexamined.

In this paper, we extend beyond prompt-level approaches and systematically investigate *concept erasure reversibility* from a parameter-level perspective. To this end, we introduce a diagnostic framework designed to assess the vulnerability of existing erasure methods. The framework leverages two minimal yet general strategies: a Gradient-Guided Probe, which restores erased behavior by reversing suppression gradients within the model parameters, and an Instance-Personalization Probe, which reinstates concepts through few-shot personalization using a small set of reference images. Complementing these empirical probes, we provide a theoretical analysis that establishes formal bounds on deviations from the original model, demonstrating that erased concepts often remain recoverable with only minor parameter updates.

We extensively evaluate the reversibility of erased concepts across six state-of-the-art erasure methods: ESD (Gandikota et al., 2023), UCE (Gandikota et al., 2024), MACE (Lu et al., 2024), FMN (Zhang et al., 2024b), AGE (Bui et al., 2025), and ConceptPrune (Chavhan et al., 2025). Our results reveal that erased concepts can often be reinstated following only minimal parameter adaptation, as confirmed by improvements across classification accuracy, CLIP alignment, and LPIPS similarity, while the model's untargeted performance remains largely unaffected. These empirical observations closely align with our theoretical analysis, which demonstrates that the reactivated model frequently approximates the original, unerased model within a bounded error. Crucially, these findings are consistent across different methods, concept categories, and diffusion backbones, indicating that recoverability is a persistent limitation of current erasure techniques and that latent representations often persist despite apparent suppression. This underscores the need for future research to develop erasure strategies that explicitly dismantle residual representations and provide stronger guarantees of irreversibility.

In summary, our contributions are as follows:

- **A diagnostic framework for reversibility.** We introduce a parameter-level framework with two lightweight probes: a Gradient-Guided Probe that restores suppressed gradients, and an Instance-Personalization Probe that rebinds concepts from a few examples. This design goes beyond prompt-based circumvention, directly evaluating whether erased concepts can be readily recovered within the model's weight space.
- **Theoretical and representation-level analysis.** We derive reactivation bounds that quantify deviations of the model's parameters from the original, unerased version. These theoretical results align with empirical measures by classification accuracy, CLIP alignment, and LPIPS perceptual similarity, providing an explanation for why erased concepts can often be recovered with only minimal adaptation.
- **Extensive empirical evaluation.** We evaluate six state-of-the-art erasure methods across both object and style concepts, using multiple diffusion backbones. Results from both probes consistently demonstrate that erased concepts can be reinstated with high fidelity following only minimal adaptation, highlighting recoverability as a fundamental limitation of current concept erasure techniques.

## 2 RELATED WORK

**Diffusion Models and Personalization.** Text-to-image diffusion models have emerged as the dominant paradigm for generative image synthesis (Rombach et al., 2022; Ramesh et al., 2022; Ho et al., 2020). Their ability to generate semantically faithful and photorealistic images has enabled widespread adoption. Personalization methods such as DreamBooth (Ruiz et al., 2023), textual inversion (Gal et al., 2022), and parameter-efficient tuning (Kumari et al., 2023; Shi et al., 2024) allow models to encode new concepts from limited data. While these techniques highlight the flexibility of diffusion models, they also amplify risks of misuse, motivating research on concept erasure (Kim & Qi, 2025; Xie et al., 2025).

**Concept Erasure in Diffusion Models.** Concept erasure aims to suppress a model's ability to generate undesired objects, styles, or identities. Representative approaches include fine-tuning (e.g., ESD (Gandikota et al., 2023)), cross-attention editing methods such as UCE and MACE (Gandikota et al., 2024; Lu et al., 2024), and attention re-steering techniques like FMN (Zhang et al., 2024b). Other directions include regularization-based methods such as RECELER (Huang et al., 2024), as

well as pruning- or adversarial-guided strategies, such as Concept Prune (Chavhan et al., 2025) and AGE (Bui et al., 2025). Recent work has also extended erasure to flow-matching architectures (Gao et al., 2025), autoregressive models (Han et al., 2025), and text-to-video models (Ye et al., 2025; Xu et al., 2025). Beyond diffusion, the literature on machine unlearning (Wang et al., 2024) shows that forgetting is inherently difficult. Studies such as RESTOR (Rezaei et al., 2024) further demonstrate that unlearned knowledge often remains recoverable. Together, these findings suggest that recoverability is a recurring challenge, motivating our parameter-level study of reversibility.

**Circumvention of Erasure.** Although research on this topic is still emerging, several recent studies have shown that erased concepts can be partially restored by optimizing or perturbing the input space. Pham et al. (2024) and Han et al. (2024) optimized adversarial embeddings via textual inversion and, in some cases, alternated with surrogate model parameter updates to recover erased concepts, achieving restoration even under black-box settings by improving transferability across unlearned models. More recently, Beerens et al. (2025) proposed RECORD, a tangential coordinate-descent algorithm that directly searches the discrete token space for seed-agnostic adversarial prompts, significantly boosting attack success rates. Lu et al. (2025) further demonstrated circumvention via paraphrased prompts, inpainting, and noise-driven probing. Complementary to these input-space studies, Rusanovsky et al. (2025) adopt a latent-space perspective, showing that high-likelihood seeds can still reconstruct erased concepts, which reinforces the view that current approaches largely suppress rather than eliminate the targeted representations and motivates exploration along other representational axes such as the parameter space. Our work follows this trajectory by directly probing the parameter space to assess whether erased concepts can be reinstated with minimal weight adaptation. Empirical results show that our probes achieve consistently high reactivation accuracy (see Appendix G), further highlighting the need for more robust and verifiable defenses, such as R.A.C.E. (Kim et al., 2024).

## 3 BEHAVIORAL REVERSIBILITY OF CONCEPT ERASURE

Concept erasure aims to suppress a model's ability to generate targeted objects, styles, or identities, facilitating safer and more controlled deployment. Although recent methods achieve effective prompt-level suppression, it remains unclear whether such erasures genuinely eliminate a model's generative capacity or merely mask it under specific inputs. We investigate this question through a three-step approach. First, we formalize the behavioral limitations of existing erasure methods in Proposition 1. Second, we introduce two lightweight parameter-level probes, namely a Gradient-Guided Probe and an Instance-Personalization Probe, designed as controlled diagnostics to determine whether erased concepts persist in latent form. Third, we provide a theoretical analysis of reactivation bounds, showing that erased concepts can often be reinstated by recovering a model that closely approximates the original, unerased version.

### 3.1 CONDITIONAL NATURE AND INTRINSIC VULNERABILITY

A generative model, parameterized by $\theta$, defines a conditional distribution $p_\theta(x \mid c)$ over images $x \in \mathcal{X}$ given prompts $c \in \mathcal{C}$. Let $\mathcal{X}_{\text{target}}$ denote the set of undesired content, and $\mathcal{C}_{\text{target}}$ denote the set of prompts explicitly targeted by an erasure method. Most existing approaches enforce a constraint of the form:

$$\forall c \in \mathcal{C}_{\text{target}}, \quad \text{supp}(p_\theta(x \mid c)) \cap \mathcal{X}_{\text{target}} = \emptyset,$$

which ensures that the model cannot generate the target concept for a restricted set of prompts. Note that in the formulation above, the constraint is defined with respect to fixed model parameters and prompts. As a result, erased concepts are often conditionally suppressed rather than fully removed, leaving latent representations that can be reactivated through slight prompt modifications or minor parameter updates. Formally, this vulnerability can be expressed as follows.

**Proposition 1** (Conditional Nature of Existing Erasure Methods). *Let $\mathcal{X}_{target} \subset \mathcal{X}$ denote a concept intended for erasure, and let $p_\theta(x \mid c)$ be the conditional distribution of a model parameterized by $\theta$. Assume an erasure algorithm enforces*

$$p_\theta(x \in \mathcal{X}_{target} \mid c) = 0, \quad \forall c \in \mathcal{C}_{target}.$$

*If there exists either (i) an arbitrarily small parameter perturbation $\delta_\theta$ or (ii) a prompt $c' \notin \mathcal{C}_{target}$ such that*

$$p_{\theta+\delta_\theta}(x \in \mathcal{X}_{target} \mid c) > 0 \quad or \quad p_\theta(x \in \mathcal{X}_{target} \mid c') > 0,$$

*then the concept has not been fundamentally erased but only conditionally suppressed, and remains recoverable under either parameter perturbations or prompt shifts.*

**Remark.** The above proposition highlights the conditional nature of many current erasure methods: they tend to suppress targeted content under specific model parameters and prompts rather than eliminating it entirely. While Proposition 1 applies universally to all models and prompts, an important question remains: how easily can the enforced erasure be undone in practice? If small parameter perturbations or slight changes in prompts are sufficient to recover the concept, then the conditional suppression implemented by current methods is highly fragile and susceptible to circumvention.

## 3.2 DIAGNOSTIC PROBES: GRADIENT-GUIDED AND INSTANCE-PERSONALIZATION

Motivated by the above observation, we now investigate the recoverability of erased concepts using multiple diagnostic probes. Specifically, we introduce two lightweight fine-tuning strategies that function as diagnostic probes: the Gradient-Guided Probe and the Instance-Personalization Probe. Both probes are deliberately minimal, serving as controlled tests to determine whether erased concepts persist in latent form. We emphasize that the probes operate as stability diagnostics rather than as training procedures. A concept that has been truly removed should resist small, lightweight updates, whereas a concept that remains dormant will reappear under minimal perturbation. Thus, successful reactivation under these deliberately weak probes provides evidence of residual structure rather than relearning.

**Gradient-Guided Probe.** This probe generalizes the idea of reversing suppression gradients. Whereas erasure methods such as ESD dampen concept-aligned gradients, the probe restores them by inverting the suppression loss into a reinforcement signal. Concretely, given a latent $x_t$ at timestep $t$, we define three embeddings: a neutral embedding $\tau(\emptyset)$ for the unconditional prompt, an anchor embedding $\tau(c^*)$ for a related but broader concept, and the target embedding $\tau(c)$ for the erased concept. For example, $c$ may be "a photo of a church" and $c^*$ "a photo of a building." We then construct a reverse-guided prediction target:

$$\epsilon_{\text{target}}(x_t, c, t) = \epsilon_\theta(x_t, c^*, t) + \gamma \cdot \left(\epsilon_\theta(x_t, c, t) - \epsilon_\theta(x_t, \emptyset, t)\right), \tag{1}$$

where $\gamma$ controls the strength of reinforcement. Lightweight adaptation updates $\theta'$ to $\theta''$ by aligning predictions with this target:

$$\mathcal{L}_{\text{Gradient-Guided}}(\theta'') = \mathbb{E}_{x_t, t}\left[\|\epsilon_{\theta''}(x_t, c, t) - \epsilon_{\text{target}}(x_t, c, t)\|^2\right]. \tag{2}$$

Here, $\theta'$ represents the erased model prior to probing, $\theta''$ represents the minimally updated model obtained after applying the probe. This procedure tests whether suppressed gradients can be reinstated with minimal effort, revealing the persistence of concept-aligned directions.

**Instance-Personalization Probe.** This probe adapts DreamBooth (Ruiz et al., 2023) for diagnostic use. Given a small reference set $\mathcal{X}_{\text{ref}} \subset \mathcal{X}_{\text{target}}$, it associates a rare token $v_*$ with the erased concept by minimizing:

$$\begin{aligned}
\mathcal{L}_{\text{Instance-Personalization}}(\theta'') = &\mathbb{E}_{x_0 \sim \mathcal{X}_{\text{ref}}, \epsilon \sim \mathcal{N}(0,I), t}\left[\|\epsilon - \epsilon_{\theta''}(z_t, t, \tau(c_{\text{inst}}))\|^2\right] \\
&+ \lambda_{\text{prior}} \mathbb{E}_{x_0 \sim \mathcal{X}_{\text{class}}, \epsilon \sim \mathcal{N}(0,I), t}\left[\|\epsilon - \epsilon_{\theta''}(z_t, t, \tau(c_{\text{class}}))\|^2\right],
\end{aligned} \tag{3}$$

where $z_t = \sqrt{\alpha_t}\, x_0 + \sqrt{1 - \alpha_t}\, \epsilon$, and $t \sim \mathcal{U}\{1, \ldots, T\}$. Here, $c_{\text{inst}}$ denotes the instance prompt (e.g., "a photo of a $v_*$"), and $c_{\text{class}}$ denotes the class prompt (e.g., "a photo of a dog"). By re-binding the erased concept to a new token using only a few reference images, this probe exposes what we term the *personalization–erasure paradox*, where erasure seeks to remove a model's ability to generate certain concepts, yet personalization methods such as DreamBooth can reinstate them with few-shot learning, even in models that have ostensibly undergone erasure. This probe operates with very limited capacity, updating only a rare token and a small subset of attention parameters. Such constrained adaptation is insufficient to synthesize a novel visual concept, and primarily serves as a diagnostic check on whether residual representations remain after erasure.

**Comparison of Probes.** Both probes operate directly at the parameter level but follow different mechanisms. The Gradient-Guided Probe reinstates suppressed concepts by performing

reverse-guided fine-tuning on the original prompts, seeking a parameter perturbation $\delta_\theta$ such that $p_{\theta+\delta_\theta}(x \in \mathcal{X}_{\text{target}} \mid c) > 0$. The Instance-Personalization Probe, in contrast, re-personalizes erased concepts using a small set of visual examples and conduct the few-shot learning. In Proposition 1, this corresponds to perturb both the parameters and prompts, i.e., $p_{\theta'}(x \in \mathcal{X}_{\text{target}} \mid c') > 0$. Together, these probes serve as minimal yet effective diagnostics that reveal the persistence of latent representations. As demonstrated in Section 4, both probes succeed with only a few fine-tuning steps, providing strong empirical evidence that current erasure methods tend to suppress rather than fully eliminate targeted concepts. As an additional sanity check, Appendix I presents a counter-example using a custom coin image that the pretrained model was never capable of generating. Even when applying the Instance-Personalization Probe with substantially more steps, the model fails to synthesize or generalize the concept. This demonstrates that the probes operate in a diagnostic regime rather than a learning regime, and therefore cannot introduce a novel visual concept that the model did not previously encode. Therefore, when erased concepts reappear after only a few lightweight probe steps, the effect reflects residual representation rather than relearning.

## 3.3 THEORETICAL ANALYSIS OF REACTIVATION BOUNDS

Building on Proposition 1, we provide a quantitative characterization of how easily erased concepts can be reinstated under parameter-level adaptation. Here we focus on the Gradient-Guided Probe, since its reverse-guided objective admits a tractable optimization form suitable for deriving convergence bounds. An embedding-level analysis for the Instance-Personalization Probe is presented in Appendix D, where we establish local ascent guarantees for token-level personalization.

We begin by modeling noisy gradient descent on the Gradient-Guided loss in the continuous-time limit (Weinan, 2017; Zhang et al., 2024a), which allows us to approximate the evolution of weight differences using a stochastic differential equation (SDE) (Arnold, 1974; Oksendal, 2013). In particular, suppose the underlying ground-truth reverse process, which corresponds to concept reactivation via flipped erasing tuning, follows

$$d\theta(t) = \nabla f(\theta(t)) \, dt + \Sigma_1(t)^{1/2} dB_1(t),$$

while the actual optimization process evolves according to

$$d\tilde{\theta}(t) = \nabla f(\tilde{\theta}(t)) \, dt + \Sigma_2(t)^{1/2} dB_2(t),$$

where $B_1(t)$ and $B_2(t)$ are independent standard $d$-dimensional Brownian motions (Einstein, 1905). We define the deviation between the two traces as

$$\delta(t) := \tilde{\theta}(t) - \theta(t).$$

The subsequent Theorems provide bounds on the weight differences under two conditions.

**Theorem 2** (General Reactivation Bound under $L$-smoothness). *Let $f : \mathbb{R}^d \to \mathbb{R}$ be $L$-smooth, i.e.,*

$$\|\nabla f(x) - \nabla f(y)\| \le L\|x - y\|, \quad \forall x, y \in \mathbb{R}^d,$$

*and assume that the noise covariances satisfy*

$$\text{tr}(\Sigma_1(t)) \le \bar{\sigma}_1, \quad \text{tr}(\Sigma_2(t)) \le \bar{\sigma}_2, \quad \forall t \ge 0.$$

*Then the expected squared deviation is bounded at the terminal time $T$:*

$$\mathbb{E}\|\delta(T)\|^2 \le \frac{\bar{\sigma}_1 + \bar{\sigma}_2}{2L} \left( e^{2LT} - 1 \right).$$

**Proof Sketch.** Using Itô's isometry (Oksendal, 2013) and $L$-smoothness, we derive the differential inequality $\frac{d}{dt} \mathbb{E}\|\delta(t)\|^2 \le 2L \, \mathbb{E}\|\delta(t)\|^2 + \mathbb{E} \, \text{tr}(\Sigma_1(t) + \Sigma_2(t))$. Applying Grönwall's inequality (Gronwall, 1919) gives the stated bound. The complete derivation is provided in Appendix B.

**Theorem 3** (Reactivation Bound under Strong Convexity). *If $f$ is additionally $\mu$-strongly convex, i.e.,*

$$\langle x - y, \nabla f(x) - \nabla f(y) \rangle \ge \mu \|x - y\|^2, \quad \forall x, y \in \mathbb{R}^d,$$

*then the expected squared deviation is bounded at the terminal time $T$:*

$$\mathbb{E}\|\delta(T)\|^2 \le \frac{\bar{\sigma}_1 + \bar{\sigma}_2}{2\mu} \left( 1 - e^{-2\mu T} \right).$$

*Thus, the expected deviation converges linearly to a noise-dependent plateau.*

**Proof Sketch.** Using strong convexity, we derive the differential inequality $\frac{d}{dt}\mathbb{E}\|\delta(t)\|^2 \leq -2\mu\,\mathbb{E}\|\delta(t)\|^2 + \mathbb{E}\operatorname{tr}(\Sigma_1(t) + \Sigma_2(t))$. Applying Grönwall's inequality yields the stated exponential decay toward the noise-dependent plateau. The complete derivation is provided in Appendix C.

**Remark.** In the general $L$-smooth case, the deviation between the two stochastic trajectories admits an upper bound, representing the worst-case scenario. By contrast, if $f$ additionally satisfies the $\mu$-strongly convex condition, the deviation $\delta(t)$ enjoys a contraction property: the expected squared difference converges to a steady-state bound, $\lim_{t\to\infty}\mathbb{E}\|\delta(t)\|^2 = (\bar{\sigma}_1 + \bar{\sigma}_2)/(2\mu)$, indicating that the system remains stable and the deviation does not accumulate even over an infinite horizon.

**Implications of Diagnostic Probes.** In practice, most neural networks are better characterized by the $L$-smooth setting in Theorem 2, where parameter deviations can, in principle, grow exponentially with the effective time horizon $T$. However, erasure methods typically employ very small learning rates (e.g., $10^{-6}$–$10^{-5}$) and perform only a limited number of tuning steps to preserve performance on untargeted concepts, keeping the effective $T$ small. Consequently, the theoretical bound on $\mathbb{E}|\delta(T)|^2$ remains modest. This observation aligns with our empirical findings: the mean absolute parameter difference after reactivation is minor (on the order of $10^{-4}$), as reported in Table 5. Furthermore, if the reverse process remains within a locally strongly convex region, the deviations may contract to a bounded region. This suggests that overcoming such contraction behavior could be necessary for designing truly irreversible erasure methods.

## 4 BEHAVIORAL PROBING OF CONCEPT ERASURE METHODS

We now turn to empirical evaluations of concept erasure and reactivation. The experiments are designed to directly address our central question: can lightweight diagnostic probes reliably reinstate concepts that state-of-the-art erasure methods claim to remove? To this end, we evaluate multiple erasure algorithms across diverse concept categories and diffusion backbones, measuring both the fidelity of reactivation and the preservation of untargeted generation quality. These empirical findings align with the stability-based interpretation developed in our diagnostic analysis (Section 5), where concepts that are only superficially suppressed reappear under minimal perturbation.

### 4.1 EXPERIMENTAL SETUP

**Backbones and Concept Classes.** Our primary experiments are conducted on Stable Diffusion v1.4 (SD1.4), the most widely used backbone in prior erasure studies. We also report validation experiments on Stable Diffusion v2.1 (SD2.1) to confirm that our findings are not specific to SD1.4. Following prior work (Gandikota et al., 2023; 2024; Lu et al., 2024), we evaluate ten ImageNet objects (cassette player, chain saw, church, gas pump, tench, garbage truck, English springer, golf ball, parachute, French horn) and five artistic styles (Pablo Picasso, Vincent van Gogh, Rembrandt, Andy Warhol, Caravaggio), ensuring both semantic diversity and comparability with existing benchmarks.

**Erasure Methods and Probes.** We benchmark six representative erasure methods: Unified Concept Editing (UCE) (Gandikota et al., 2024), Erased Stable Diffusion (ESD) (Gandikota et al., 2023), MACE (Lu et al., 2024), FMN (Zhang et al., 2024b), AGE (Bui et al., 2025), and ConceptPrune (Chavhan et al., 2025), covering projection, fine-tuning, adversarial training, and pruning paradigms. To test reversibility, we apply two diagnostic probes: the Gradient-Guided Probe, which restores erased concepts via lightweight gradient reversal, and the Instance-Personalization Probe, which reinstates concepts through few-shot personalization with a small reference set.

**Metrics and Evaluation Protocol.** We evaluate along two axes: (i) **Reactivation Accuracy**, measured as Top-1 classification accuracy using a pretrained ResNet-50 (He et al., 2016) for object concepts, and (ii) **Generative Quality**, assessed using CLIP similarity (Radford et al., 2021) for style concepts and LPIPS perceptual distance (Zhang et al., 2018) with AlexNet (Krizhevsky et al., 2012). Following prior work (Gandikota et al., 2023; 2024), we use a predefined prompt list, with 10 prompts per class and 20 images per prompt at $512 \times 512$ resolution, yielding 200 images per class (2,000 object images and 1,000 style images in total).

**Implementation Details.** For the Gradient-Guided Probe, we fine-tune the UNet attention modules for 200 steps using Adam. For object concepts, we set the learning rate to $1 \times 10^{-5}$ and the erasure/reactivation strength to 0.8; for artistic styles, we adopt $5 \times 10^{-5}$ and a strength of 10.0. For the Instance-Personalization Probe, we fine-tune the UNet backbone (and optionally the text encoder) for 500 steps using Adam with a learning rate of $1 \times 10^{-6}$, employing both instance and class

Table 1: Evaluation results on ten object classes with Gradient-Guided and Instance-Personalization Probes. Each cell reports "**Erased** (↓) **/ Reactivation** (↑)", where lower values indicate stronger erasure and higher values indicate more successful reactivation. All values are Top-1 classification accuracies measured by a pretrained ResNet-50.

(a) Gradient-Guided Probe

| Object | Original | ESD | UCE | MACE | FMN | AGE | CP |
|---|---|---|---|---|---|---|---|
| cassette player | 74.0 | 0.5 / 70.0 | 3.5 / 22.5 | 21.5 / 78.0 | 5.5 / 46.5 | 12.0 / 65.0 | 53.5 / 85.5 |
| chain saw | 78.0 | 0.0 / 86.0 | 0.0 / 29.5 | 1.0 / 45.0 | 0.0 / 75.5 | 5.0 / 83.5 | 15.0 / 56.0 |
| church | 86.5 | 7.0 / 93.5 | 23.0 / 83.0 | 10.0 / 84.0 | 6.5 / 94.5 | 65.5 / 87.0 | 73.0 / 93.5 |
| english springer | 93.0 | 0.0 / 81.5 | 0.5 / 72.5 | 9.0 / 81.5 | 0.0 / 8.5 | 4.5 / 89.0 | 24.0 / 44.5 |
| french horn | 100.0 | 0.0 / 99.5 | 2.5 / 99.0 | 16.0 /100.0 | 21.0 / 97.5 | 29.5 / 99.5 | 34.5 /100.0 |
| garbage truck | 82.0 | 7.5 / 81.0 | 7.5 / 22.5 | 1.5 / 47.5 | 0.0 / 5.0 | 0.0 / 71.5 | 3.5 / 59.0 |
| gas pump | 73.5 | 0.0 / 60.5 | 5.0 / 28.5 | 16.0 / 53.5 | 1.5 / 74.5 | 11.0 / 70.5 | 54.0 / 63.0 |
| tench | 75.0 | 0.0 / 54.0 | 0.5 / 0.5 | 38.0 / 62.0 | 0.0 / 14.5 | 1.5 / 61.0 | 0.5 / 59.5 |
| golf ball | 98.5 | 0.0 / 97.0 | 65.5 / 91.0 | 2.0 / 78.5 | 18.5 / 96.5 | 20.0 / 97.0 | 81.0 / 99.0 |
| parachute | 93.0 | 0.0 / 95.5 | 9.5 / 49.5 | 49.0 / 87.5 | 1.5 / 80.5 | 8.5 / 87.5 | 4.0 / 87.0 |
| Average | 88.85 | 1.5 / 81.85 | 11.75 / 49.85 | 16.4 / 71.75 | 5.45 / 59.35 | 15.75 / 81.15 | 34.3 / 74.7 |

(b) Instance-Personalization Probe

| Object | Original | ESD | UCE | MACE | FMN | AGE | CP |
|---|---|---|---|---|---|---|---|
| cassette player | 74.0 | 0.5 / 51.5 | 3.5 / 15.5 | 21.5 / 47.5 | 5.5 / 31.5 | 12.0 / 61.0 | 53.5 / 54.5 |
| chain saw | 78.0 | 0.0 / 75.0 | 0.0 / 68.0 | 1.0 / 82.0 | 0.0 / 63.5 | 5.0 / 66.0 | 15.0 / 40.5 |
| church | 86.5 | 7.0 / 98.0 | 23.0 / 98.0 | 10.0 / 96.0 | 6.5 / 94.5 | 65.5 / 94.5 | 73.0 / 91.5 |
| english springer | 93.0 | 0.0 / 87.5 | 0.5 / 97.0 | 9.0 / 83.0 | 0.0 / 99.0 | 4.5 / 97.0 | 24.0 / 99.0 |
| french horn | 100.0 | 0.0 / 99.5 | 2.5 /100.0 | 16.0 /100.0 | 21.0 /100.0 | 29.5 /100.0 | 34.5 / 99.0 |
| garbage truck | 82.0 | 7.5 / 61.0 | 7.5 / 89.0 | 1.5 / 72.5 | 0.0 / 66.5 | 0.0 / 63.0 | 3.5 / 84.5 |
| gas pump | 73.5 | 0.0 / 29.5 | 5.0 / 48.5 | 16.0 / 59.5 | 1.5 / 80.5 | 11.0 / 72.5 | 54.0 / 82.5 |
| tench | 75.0 | 0.0 / 42.5 | 0.5 / 7.0 | 38.0 / 72.5 | 0.0 / 95.5 | 1.5 / 71.0 | 0.5 / 71.5 |
| golf ball | 98.5 | 0.0 / 96.5 | 65.5 / 90.5 | 2.0 / 92.5 | 18.5 / 80.5 | 20.0 /100.0 | 81.0 / 97.5 |
| parachute | 93.0 | 0.0 / 78.0 | 9.5 / 80.0 | 49.0 / 88.5 | 1.5 /100.0 | 8.5 / 90.5 | 4.0 / 74.5 |
| Average | 88.85 | 1.5 / 71.9 | 11.75 / 69.35 | 16.4 / 79.4 | 5.45 / 81.65 | 15.75 / 81.55 | 34.3 / 79.5 |

prompts, with prior preservation regularization $\lambda_{\mathrm{prior}} = 1.0$ to mitigate overfitting. All experiments are run on a single NVIDIA RTX 4090 GPU with 24GB memory.

### 4.2 ANALYSIS OF EXPERIMENTAL RESULTS

#### 4.2.1 ERASURE REVERSIBILITY ON OBJECT CONCEPTS

As shown in Table 1, among the six methods, ESD, FMN, and AGE achieve the strongest suppression, often driving erased accuracies close to zero across multiple classes (e.g., chain saw, tench, and French horn). Yet both the Gradient-Guided Probe and the Instance-Personalization Probe rapidly reinstate these concepts, with recovery levels approaching or even matching the original model (e.g., French horn and golf ball restored to nearly 100%). Projection-based UCE suppresses less aggressively, leaving substantial residual signals (e.g., golf ball at 65.5%), which the Instance-Personalization Probe can almost completely reinstate (e.g., church rising from 23% to 98%). Similarly, pruning-based CP occasionally achieves strong erasure (e.g., tench reduced to 0.5%) but is easily reversed by personalization (e.g., English springer restored from 24% to 99%). Overall, these results indicate that while methods differ in suppression strength, all leave recoverable traces in latent space, confirming that current erasure strategies achieve only conditional suppression rather than irreversible removal of object concepts. Visual comparison of object-level erasure and reactivation is provided in Appendix F.1.

#### 4.2.2 ERASURE REVERSIBILITY ON ARTISTIC STYLES

Table 2 summarizes the outcomes of six erasure methods across five artist styles. We observe that all approaches achieve some degree of suppression, with ESD and AGE producing the largest initial reductions in CLIP similarity (e.g., Van Gogh suppressed to 19.78, Caravaggio to 18.53). However, despite these apparent gains, both the Gradient-Guided Probe and the Instance-Personalization Probe consistently restore erased styles to near-original levels. For instance, Van Gogh, which drops to 19.78 under ESD, returns to 30.07 after Instance-Personalization reactivation, nearly matching the original 30.30. The two probes reveal different vulnerabilities: the Gradient-Guided Probe effectively reverses gradient-based methods such as MACE and FMN, while the Instance-Personalization Probe excels at reinstating styles under pruning- and adversarial-guided erasures such as CP and AGE. Table 3 reports LPIPS distances, providing a perceptual measure of similarity to the original model. Consistent with the CLIP results, erasure typically increases LPIPS substantially, whereas

Table 2: Comparison of six erasure methods on five artist-style concepts. Each cell reports "**Erased** (↓) / **Reactivation** (↑)", where lower values indicate stronger erasure and higher values indicate more successful reactivation. "Original" denotes CLIP similarity of the unmodified model.

(a) Gradient-Guided Probe

| Artist | Original | ESD | UCE | MACE | FMN | AGE | CP |
|---|---|---|---|---|---|---|---|
| Andy Warhol | 30.07 | 22.44 / 30.65 | 24.24 / 29.68 | 24.61 / 29.14 | 24.10 / 29.74 | 20.66 / 27.23 | 21.63 / 26.19 |
| Pablo Picasso | 28.75 | 23.17 / 28.59 | 25.53 / 26.90 | 25.92 / 27.85 | 23.85 / 27.51 | 21.62 / 26.86 | 21.16 / 26.11 |
| Van Gogh | 30.30 | 19.78 / 29.73 | 25.45 / 29.56 | 25.66 / 29.94 | 25.93 / 27.96 | 19.08 / 27.56 | 20.59 / 28.52 |
| Rembrandt | 29.45 | 20.41 / 28.94 | 24.17 / 28.91 | 26.01 / 29.05 | 26.50 / 27.87 | 20.02 / 27.80 | 22.58 / 27.07 |
| Caravaggio | 28.46 | 19.01 / 27.33 | 23.45 / 27.11 | 24.16 / 27.63 | 25.01 / 29.16 | 18.53 / 27.77 | 20.65 / 27.88 |

(b) Instance-Personalization Probe

| Artist | Original | ESD | UCE | MACE | FMN | AGE | CP |
|---|---|---|---|---|---|---|---|
| Andy Warhol | 30.07 | 22.44 / 28.55 | 24.24 / 27.38 | 24.61 / 30.07 | 24.10 / 26.64 | 20.66 / 27.04 | 21.63 / 29.24 |
| Pablo Picasso | 28.75 | 23.17 / 29.01 | 25.53 / 27.56 | 25.92 / 25.59 | 23.85 / 24.05 | 21.62 / 27.18 | 21.16 / 27.75 |
| Van Gogh | 30.30 | 19.78 / 30.07 | 25.45 / 28.90 | 25.66 / 27.26 | 25.93 / 26.89 | 19.08 / 28.28 | 20.59 / 30.68 |
| Rembrandt | 29.45 | 20.41 / 26.21 | 24.17 / 27.42 | 26.01 / 26.05 | 26.50 / 28.75 | 20.02 / 25.98 | 22.58 / 29.42 |
| Caravaggio | 28.46 | 19.01 / 26.63 | 23.45 / 27.11 | 24.16 / 25.33 | 25.01 / 28.54 | 18.53 / 27.03 | 20.65 / 28.00 |

Table 3: LPIPS comparison for erasure and reactivation on five artist-style concepts (lower is better). Each cell reports "**Erased** (↑) / **Gradient-Guided Probe** (↓) / **Instance-Personalization Probe** (↓)". Higher values in the first entry indicate stronger erasure, while lower values in the latter two entries indicate more successful reactivation. LPIPS is computed between images generated by erased/reactivated models and those from the original model.

| Artist | ESD | UCE | MACE | FMN | AGE | CP |
|---|---|---|---|---|---|---|
| Andy Warhol | 0.88 / 0.42 / 0.61 | 0.65 / 0.50 / 0.59 | 0.78 / 0.49 / 0.65 | 0.77 / 0.51 / 0.61 | 0.84 / 0.53 / 0.71 | 0.73 / 0.62 / 0.65 |
| Pablo Picasso | 0.82 / 0.27 / 0.55 | 0.45 / 0.40 / 0.47 | 0.53 / 0.38 / 0.53 | 1.11 / 0.44 / 0.63 | 0.71 / 0.44 / 0.58 | 0.81 / 0.58 / 0.63 |
| Van Gogh | 0.83 / 0.33 / 0.53 | 0.54 / 0.41 / 0.49 | 0.57 / 0.45 / 0.53 | 0.84 / 0.49 / 0.59 | 0.77 / 0.46 / 0.59 | 0.67 / 0.61 / 0.57 |
| Rembrandt | 0.90 / 0.37 / 0.67 | 0.55 / 0.40 / 0.58 | 0.60 / 0.39 / 0.51 | 0.81 / 0.47 / 0.51 | 0.78 / 0.50 / 0.68 | 0.78 / 0.71 / 0.65 |
| Caravaggio | 0.90 / 0.31 / 0.57 | 0.49 / 0.35 / 0.47 | 0.48 / 0.38 / 0.45 | 0.69 / 0.43 / 0.52 | 0.80 / 0.39 / 0.64 | 0.73 / 0.54 / 0.59 |

both probes consistently reduce these values. This confirms that reactivated models not only recover semantic alignment (via CLIP) but also regain perceptual fidelity (via LPIPS), reinforcing the conclusion that current erasure techniques achieve only conditional suppression, leaving the underlying stylistic representations intact and readily recoverable under minimal fine-tuning. Visual comparison of style-level erasure and reactivation is provided in Appendix F.2.

### 4.3 GENERALIZATION ACROSS BACKBONES AND RESOLUTIONS

We conducted additional experiments on Stable Diffusion 2.1 and Stable Diffusion 1.4 at a lower input resolution of $256 \times 256$ to examine whether the vulnerabilities of concept erasure generalize across model backbones and resolutions. As shown in Table 4, strong suppression is consistently achieved, often driving erased accuracies close to zero. However, once subjected to lightweight reactivation, the erased concepts are almost fully restored, with accuracies nearly matching those of the original models. These findings indicate that recoverability of erased concepts is not tied to a particular erasure method, backbone, or resolution, but instead reflects an inherent limitation common to current erasure approaches.

### 4.4 REACTIVATION DYNAMICS UNDER PARAMETER PERTURBATION

To further investigate the relationship between reactivation performance and parameter updates, we analyzed reactivation iterations for two representative object classes (chain saw and tench). As shown in Table 5, increasing the number of fine-tuning iterations consistently improves the accuracy of the erased class while requiring only lightweight parameter changes. For example, for *tench*, 20 iterations recover 32.0% accuracy with just 0.34% of parameters modified, whereas 50 iterations achieve 61.0% with 0.93% modified, and 200 iterations reach 66.0% with 1.84%. A similar trend is observed for *chain saw*, where accuracy rises from 19.5% to 81.0% as iterations increase from 20 to 200, with less than 1.4% of parameters changed. Importantly, the accuracy drop compared to the original unerased model on non-target classes remains small (3–5%) across all settings, indicating limited side effects. These results demonstrate that erased concepts can be efficiently reactivated with modest weight perturbations, reinforcing that current erasure methods provide only superficial suppression rather than permanent removal.

Table 4: Evaluation of reactivation across model versions and resolutions. Each cell reports "Erased / Reactivation" Top-1 accuracies (ResNet-50). Results demonstrate that the vulnerabilities of concept erasure generalize across both Stable Diffusion 2.1 (a) and Stable Diffusion 1.4 at $256 \times 256$ (b).

(a) Stable Diffusion 2.1

| Object | Original | Erasure / Reactivation |
|---|---|---|
| cassette player | 63.5 | 0.5 / 54.5 |
| chain saw | 97.5 | 0.0 / 91.0 |
| church | 99.0 | 71.5 / 97.0 |
| english springer | 98.0 | 0.0 / 95.5 |
| french horn | 83.0 | 0.0 / 87.5 |
| garbage truck | 65.5 | 0.5 / 60.0 |
| gas pump | 98.0 | 0.0 / 94.5 |
| tench | 91.0 | 0.5 / 91.5 |
| golf ball | 91.0 | 1.5 / 94.0 |
| parachute | 86.0 | 0.5 / 86.0 |

(b) Stable Diffusion 1.4 ($256 \times 256$)

| Object | Original | Erasure / Reactivation |
|---|---|---|
| cassette player | 67.50 | 0.00 / 9.75 |
| chain saw | 77.25 | 0.00 / 78.25 |
| church | 77.00 | 4.75 / 74.50 |
| english springer | 89.25 | 0.00 / 89.75 |
| french horn | 99.75 | 0.00 / 100.00 |
| garbage truck | 78.75 | 0.00 / 77.75 |
| gas pump | 65.75 | 0.00 / 80.00 |
| golf ball | 93.25 | 0.00 / 95.25 |
| parachute | 93.00 | 0.00 / 89.75 |
| tench | 68.00 | 0.00 / 71.95 |

Table 5: Reactivation accuracy and weight perturbation under ESD erasure for two representative classes. Reactivation accuracy increases with iterations while parameter changes remain modest, demonstrating the fragility of ESD-based erasure.

| Iterations | Class | Original Acc | Reactivated Acc ↑ | Non-target Drop ↓ | Params Updated | Mean Abs. Change |
|---|---|---|---|---|---|---|
| 20 | chain saw | 78.0 | 19.5 | 3.7 | 0.30 | $2.29 \times 10^{-4}$ |
| 50 | chain saw | 78.0 | 80.5 | 2.9 | 0.60 | $2.53 \times 10^{-4}$ |
| 200 | chain saw | 78.0 | 81.0 | 3.0 | 1.38 | $3.06 \times 10^{-4}$ |
| 20 | tench | 75.0 | 32.0 | 5.0 | 0.34 | $2.30 \times 10^{-4}$ |
| 50 | tench | 75.0 | 61.0 | 5.3 | 0.93 | $3.08 \times 10^{-4}$ |
| 200 | tench | 75.0 | 66.0 | 3.2 | 1.84 | $3.34 \times 10^{-4}$ |

## 4.5 UNTARGETED IMPACT AND COLLATERAL EFFECTS

We further examined the untargeted effects of erasure and reactivation on Stable Diffusion 2.1. As shown in Table 6, while erasure reduces the accuracy of non-target classes (e.g., a 27.3% drop for *chain saw*), reactivation largely restores their performance, typically within 2–3% of the original accuracy. This result indicates that reactivation does not substantially disrupt untargeted classes, and confirms that our main findings generalize across models and do not arise from spurious artifacts.

## 4.6 RUNTIME OF REACTIVATION STRATEGIES

A practical concern is the computational cost of reactivating erased concepts. Across all settings, we observe that reactivation is remarkably efficient: all procedures complete under seven minutes on a single NVIDIA RTX 4090 GPU. To quantify this, we measured wall-clock runtimes on Stable Diffusion 1.4 at $256 \times 256$ resolution for a representative object class (*cassette player*) under four erasure–reactivation configurations: ESD-Erase with the Gradient-Guided Probe, ESD-Erase with the Instance-Personalization Probe, UCE-Erase with the Gradient-Guided Probe, and UCE-Erase with the Instance-Personalization Probe. Each configuration was repeated three times. On average, the Instance-Personalization Probe required about 245 seconds, while the Gradient-Guided Probe required about 369 seconds.

## 4.7 IMPLICATIONS FOR CONCEPT ERASURE

Our findings suggest that current erasure methods suffer from a fundamental weakness: they suppress the target concept at the prompt level rather than eliminating it from the parameter space, thus functioning as input filtering rather than genuine erasure. This interpretation is consistent with prior works (Pham et al., 2024; Lu et al., 2025), which show that underlying information persists in the model and can be readily recovered by optimizing special embeddings. Our results provide further evidence: as shown in Table 5, successful reactivation requires only modest parameter adjustments (typically less than 2% of weights). The small magnitude of change needed to recover erased concepts strongly indicates that the erased information is only weakly suppressed rather than globally eliminated. Addressing this limitation may require more robust defenses, such as structural interventions or alignment-driven regularization strategies, to achieve truly irreversible concept erasure.

Table 6: Target and untargeted impact of erasure and reactivation on Stable Diffusion 2.1 across ten object concepts. Each cell reports Top-1 accuracy measured by a pretrained ResNet-50. "Target" columns show performance on the erased class, while "Untargeted" columns report the performance over the remaining classes.

| Object | Target | | | Untargeted | | | |
|---|---|---|---|---|---|---|---|
| | Original | Erased ↓ | Reactivation ↑ | Original | Erased ↑ | Drop ↓ | Reactivation ↑ |
| cassette player | 63.5 | 0.5 | 54.5 | 89.9 | 79.3 | 10.6 | 86.7 |
| chain saw | 97.5 | 0.0 | 91.0 | 86.1 | 58.8 | 27.3 | 83.4 |
| church | 99.0 | 71.5 | 97.0 | 86.0 | 84.9 | 1.1 | 84.7 |
| english springer | 98.0 | 0.0 | 95.5 | 86.0 | 67.6 | 18.4 | 83.6 |
| french horn | 83.0 | 0.0 | 87.5 | 87.7 | 73.3 | 14.4 | 85.0 |
| garbage truck | 65.5 | 0.5 | 60.0 | 89.7 | 72.8 | 16.9 | 87.4 |
| gas pump | 98.0 | 0.0 | 94.5 | 86.1 | 69.7 | 16.4 | 82.4 |
| tench | 91.0 | 0.5 | 91.5 | 86.8 | 76.6 | 10.2 | 84.8 |
| golf ball | 91.0 | 1.5 | 94.0 | 86.8 | 78.6 | 8.2 | 84.7 |
| parachute | 86.0 | 0.5 | 86.0 | 87.4 | 81.3 | 6.1 | 85.4 |

## 5 From Concept Erasure to True Absence: Understanding Through Stability

Since a sufficiently large amount of fine-tuning can, in principle, teach a model nearly any concept, the key question is not whether the model can eventually relearn the concept, but how easily the concept can reappear under small, lightweight adjustments.

A useful way to interpret this distinction is through the classical physics notion of equilibrium. Physical systems exhibit multiple types of equilibrium—most notably stable and unstable states. In a stable equilibrium, small perturbations dissipate and the system returns to its original configuration. In an unstable equilibrium, even an infinitesimal disturbance pushes the system away rapidly.

The same principle applies to a model's behavior after concept erasure. A model placed into a genuinely stable erased state should resist small optimization steps: few-shot personalization or light-tuning should not be sufficient to reinstate the erased concept. In contrast, if only a handful of gradient steps reliably bring back the concept, then the erased state behaves like an unstable or weakly metastable equilibrium, superficially altered but retaining latent, easily reactivated structure.

To highlight this distinction, we examine a counter-example involving a custom coin image (Figure 4). The original Stable Diffusion model cannot generate this coin at all, indicating that it lacks a meaningful internal representation of the concept. Even after applying our Instance-Personalization Probe with a small set of ground-truth images, the model still fails to synthesize convincing images. When the model has never possessed a concept, small perturbations simply cannot inject it: the resulting generations lack fidelity and fail to generalize.

This behavior contrasts sharply with what we observe for erased concepts. Across multiple erasure methods, concepts that are claimed to be removed consistently reappear after only a few probe steps, far too rapidly to be the result of learning from scratch. Such quick recovery demonstrates that residual representational traces remain in the model, making the erasure analogous to an unstable equilibrium. Overall, this analysis emphasizes that complete erasure should make reintroduction no easier than teaching a truly novel concept, requiring substantial optimization rather than trivial adjustments.

## 6 Conclusion

We presented a systematic parameter-level study of concept erasure in diffusion models. Using two lightweight diagnostic probes, namely a Gradient-Guided Probe and an Instance-Personalization Probe, our theoretical and empirical results show that current erasure methods often achieve conditional suppression rather than complete elimination. Erased concepts can be reinstated by adapting fewer than $2\%$ of model parameters, with minimal impact on untargeted content, highlighting recoverability as a key open challenge. Future work should aim for methods that more effectively target residual representations and offer verifiable guarantees of irreversibility, potentially drawing on techniques from machine unlearning, watermarking, and model alignment to enable safer deployment of generative AI.

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

# 7 APPENDIX

## A JOINT PERTURBATION EXISTENCE (PARAMETERS AND PROMPTS)

Proposition 1 characterizes scenarios in which perturbing either the model parameters or the prompts can make the probability of generating the target concept nonzero. Building on this, we consider the case of perturbing both parameters and prompts simultaneously.

**Proposition 4** (Conditional Nature of Existing Erasure Methods)**.** *Let $\mathcal{X}_{target} \subset \mathcal{X}$ denote a concept intended for erasure, and let $p_\theta(x \mid c)$ be the conditional distribution of a model parameterized by $\theta$. Assume an erasure algorithm enforces*

$$p_\theta(x \in \mathcal{X}_{target} \mid c) = 0, \quad \forall\, c \in \mathcal{C}_{target}.$$

*If there exist arbitrarily small perturbations $\delta_\theta$ of the parameters and a prompt $c' \notin \mathcal{C}_{target}$ such that for $\theta' = \theta + \delta_\theta$,*

$$p_{\theta'}(x \in \mathcal{X}_{target} \mid c') > 0,$$

*then the concept has not been fundamentally erased but only conditionally suppressed, and remains recoverable through such joint interventions.*

**Remark.** The above proposition highlights the conditional nature of existing erasure methods: they suppress targeted content only under specific prompts rather than eliminating it entirely. Consequently, erased concepts may remain dormant within the model, leaving the possibility of recovery through either minor parameter perturbations or small prompt variations.

## B PROOF OF THEOREM 2

For completeness, we provide the full proof of Theorem 2, which in the main text was only summarized as a sketch.

**Proof.** We first make explicit the standard assumptions used in the bound.

*Assumptions.* (i) $f$ has an $L$-Lipschitz gradient in a neighborhood containing the traces, that is

$$\|\nabla f(u) - \nabla f(v)\| \le L\|u - v\| \quad \forall u, v \in \mathbb{R}^d,$$

(ii) The reference and actual dynamics start from the same point: $\theta(0) = \tilde{\theta}(0) = \theta_0$. (iii) The two stochastic processes follow the Itô SDEs

$$d\theta(t) = \nabla f(\theta(t))\, dt + \Sigma_1(t)^{1/2} dB_1(t), \qquad d\tilde{\theta}(t) = \nabla f(\tilde{\theta}(t))\, dt + \Sigma_2(t)^{1/2} dB_2(t),$$

where $B_1(t), B_2(t)$ are independent standard $d$-dimensional Brownian motions, and $\mathrm{tr}(\Sigma_1(t)) \le \bar{\sigma}_1$, $\mathrm{tr}(\Sigma_2(t)) \le \bar{\sigma}_2$ for all $t \ge 0$. Define the deviation $\delta(t) := \tilde{\theta}(t) - \theta(t)$ so that $\delta(0) = 0$.

*Step 1: Itô formula for the squared norm.* Consider $V(\delta) = \|\delta\|^2$. By Itô's formula,

$$dV(\delta(t)) = 2\langle \delta(t), d\delta(t)\rangle + \mathrm{tr}(\Sigma_1(t) + \Sigma_2(t))\, dt.$$

Since

$$d\delta(t) = d\tilde{\theta}(t) - d\theta(t) = \big(\nabla f(\tilde{\theta}(t)) - \nabla f(\theta(t))\big)\, dt + \Sigma_2(t)^{1/2} dB_2(t) - \Sigma_1(t)^{1/2} dB_1(t),$$

we have

$$d\|\delta(t)\|^2 = 2\langle \delta(t), \nabla f(\tilde{\theta}(t)) - \nabla f(\theta(t))\rangle dt + 2\langle \delta(t), dW(t)\rangle + \mathrm{tr}(\Sigma_1(t) + \Sigma_2(t))dt,$$

where $dW(t) := \Sigma_2(t)^{1/2} dB_2(t) - \Sigma_1(t)^{1/2} dB_1(t)$.

*Step 2: Take expectations.* The stochastic integral term has zero expectation, hence

$$\frac{d}{dt}\mathbb{E}\|\delta(t)\|^2 = 2\,\mathbb{E}\langle \delta(t), \nabla f(\tilde{\theta}(t)) - \nabla f(\theta(t))\rangle + \mathbb{E}\,\mathrm{tr}(\Sigma_1(t) + \Sigma_2(t)).$$

*Step 3: Use $L$-smoothness to bound the drift term.* By $L$-smoothness,

$$\|\nabla f(\tilde{\theta}(t)) - \nabla f(\theta(t))\| \le L\|\delta(t)\|,$$

so that

$$2\langle\delta(t),\nabla f(\tilde{\theta}(t))-\nabla f(\theta(t))\rangle\leq 2L\|\delta(t)\|^2.$$

Using the bound on the trace of the noise covariance gives

$$\frac{d}{dt}\mathbb{E}\|\delta(t)\|^2\leq 2L\mathbb{E}\|\delta(t)\|^2+\bar{\sigma}_1+\bar{\sigma}_2.$$

*Step 4: Solve the differential inequality.* Let $y(t):=\mathbb{E}\|\delta(t)\|^2$. Then

$$y'(t)\leq 2Ly(t)+\bar{\sigma}_1+\bar{\sigma}_2,\quad y(0)=0.$$

By the integrating factor method or Grönwall's inequality,

$$y(t)\leq\frac{\bar{\sigma}_1+\bar{\sigma}_2}{2L}\left(e^{2Lt}-1\right).$$

Setting $t$ to the desired time completes the proof:

$$\mathbb{E}\|\delta(t)\|^2\leq\frac{\bar{\sigma}_1+\bar{\sigma}_2}{2L}\left(e^{2Lt}-1\right).$$

$\square$

## C  PROOF OF THEOREM 3

**Proof.** The first two steps proceed in the same way as in the proof of Theorem 2 (see Appendix B). The key difference arises in Steps 3 and 4, where the $\mu$-PL condition allows us to establish a contraction bound rather than an expansion bound.

*Step 3: Use strongly convex condition to bound the drift term.* By the $\mu$-strongly convex condition,

$$\langle\delta(t),\nabla f(\tilde{\theta}(t))-\nabla f(\theta(t))\rangle\geq\mu\|\delta(t)\|^2.$$

Using the bounds on the trace gives the differential inequality

$$\frac{d}{dt}\mathbb{E}\|\delta(t)\|^2\leq-2\mu\mathbb{E}\|\delta(t)\|^2+\bar{\sigma}_1+\bar{\sigma}_2.$$

*Step 4: Solve the differential inequality.* Let $y(t)=\mathbb{E}\|\delta(t)\|^2$. Then

$$y'(t)\leq-2\mu y(t)+\bar{\sigma}_1+\bar{\sigma}_2,\quad y(0)=0.$$

Solving gives

$$y(t)\leq\frac{\bar{\sigma}_1+\bar{\sigma}_2}{2\mu}\left(1-e^{-2\mu t}\right),$$

which completes the proof. $\square$

## D   ANALYSIS FOR INSTANCE-PERSONALIZATION PROBE

**Setup and Notation.** Let $\theta$ denote model parameters and $\tau(\cdot)$ the text encoder that maps prompts to embeddings. DreamBooth introduces a rare token $v_*$ with embedding $e := \tau(v_*)$ and uses two prompts: $c_{\text{inst}}$ for the instance prompt such as "a photo of $v_*$" and $c_{\text{class}}$ for the class prompt such as "a photo of a dog". Let $x_0 \in \mathcal{X}_{\text{ref}}$ be a reference image for the erased concept and let $z_t = \sqrt{\alpha_t}\, x_0 + \sqrt{1-\alpha_t}\, \epsilon$ with $t$ sampled from a predefined schedule and $\epsilon \sim \mathcal{N}(0, I)$. The DreamBooth loss is

$$\mathcal{L}_{\text{Instance-Personalization}}(\theta, e) = \mathbb{E}\big[\|\epsilon - \epsilon_\theta(z_t, t, \tau(c_{\text{inst}}(e)))\|^2\big] + \lambda_{\text{prior}}\, \mathbb{E}\big[\|\epsilon - \epsilon_\theta(z_t, t, \tau(c_{\text{class}}))\|^2\big].$$

In reactivation we mainly optimize $e$ and optionally a small subset of $\theta$. For analysis it is convenient to work with a differentiable surrogate score $s_\theta(e)$ that increases when the erased concept is better reconstructed (e.g., a CLIP similarity with the concept text, or the negative instance loss).

We provide the local nondegeneracy statement in the token embedding space and provide a quantitative ascent guarantee.

**Proposition 5** (Quantitative local ascent in the token embedding). *Let $s_\theta : \mathbb{R}^{d_e} \to \mathbb{R}$ be differentiable. Assume that $\nabla_e s_\theta$ is $L_e$-Lipschitz in a neighborhood of $e_0$ and $\nabla_e s_\theta(e_0) \neq 0$. Let $u = \nabla_e s_\theta(e_0)/\|\nabla_e s_\theta(e_0)\|$ and $e_1 = e_0 + \eta u$. Then for any step size $\eta$ in the open interval $\big(0, 2\|\nabla_e s_\theta(e_0)\|/L_e\big)$ one has*

$$s_\theta(e_1) \;\geq\; s_\theta(e_0) \;+\; \eta\,\|\nabla_e s_\theta(e_0)\| \;-\; \tfrac{L_e}{2}\,\eta^2 \;>\; s_\theta(e_0).$$

*In particular, the choice $\eta^\star = \|\nabla_e s_\theta(e_0)\|/L_e$ maximizes the right-hand side and yields*

$$s_\theta(e_0 + \eta^\star u) \;\geq\; s_\theta(e_0) \;+\; \frac{\|\nabla_e s_\theta(e_0)\|^2}{2L_e}.$$

*Proof.* By $L_e$-smoothness of $s_\theta$,

$$s_\theta(e_0 + \Delta) \;\geq\; s_\theta(e_0) \;+\; \langle \nabla_e s_\theta(e_0),\, \Delta \rangle \;-\; \frac{L_e}{2}\,\|\Delta\|^2 \quad \text{for all } \Delta.$$

Taking $\Delta = \eta u$ with $u$ the normalized gradient direction gives the bound. The quadratic is positive on $\big(0, 2\|\nabla_e s_\theta(e_0)\|/L_e\big)$ and is maximized at $\eta^\star = \|\nabla_e s_\theta(e_0)\|/L_e$. $\qquad\square$

**Proposition 6** (Second-order ascent at a first-order critical point). *Assume $\nabla_e s_\theta(e_0) = 0$ and there exists a unit vector $v$ with $v^\top \nabla^2_{ee} s_\theta(e_0)\, v > 0$. Then for sufficiently small $\eta > 0$,*

$$s_\theta(e_0 + \eta v) \;=\; s_\theta(e_0) \;+\; \tfrac{1}{2}\eta^2\, v^\top \nabla^2_{ee} s_\theta(e_0)\, v \;+\; o(\eta^2) \;>\; s_\theta(e_0).$$

*Hence even at a first-order stationary point, a nondegenerate positive-curvature direction yields a local increase and initiates reactivation.* $\qquad\square$

Therefore, it is always possible to construct an embedding $\tau(c_{\text{inst}})$ along this direction, or via continuous optimization methods such as textual inversion, that maximizes the likelihood of the concept.

Moreover, in instance-personalization probes, the parameters $\theta$ can always be further optimized to improve performance, analogous to Proposition 5 6. We omit the details here for simplicity.

## E   STATISTICAL ROBUSTNESS OF REACTIVATION RESULTS

To assess the stability of our findings, we repeated the full reactivation experiment five times with different random seeds under the ESD erasure setting combined with the Gradient-Guided Probe for reactivation. For example, for the *chain saw* class, the mean reactivation accuracy was $75.0\%$ with a standard deviation of $2.62\%$, and for *tench* it was $64.6\%$ with a standard deviation of $2.88\%$. Similar relatively small variance was observed across other categories, suggesting that the reactivation results are generally stable and reproducible.

## F   QUALITATIVE VISUALIZATION.

In addition to quantitative metrics, we present qualitative visualizations to highlight the effectiveness of our diagnostic probes. These examples focus on how erased concepts can be reinstated with high fidelity, illustrating the persistence of residual representations.

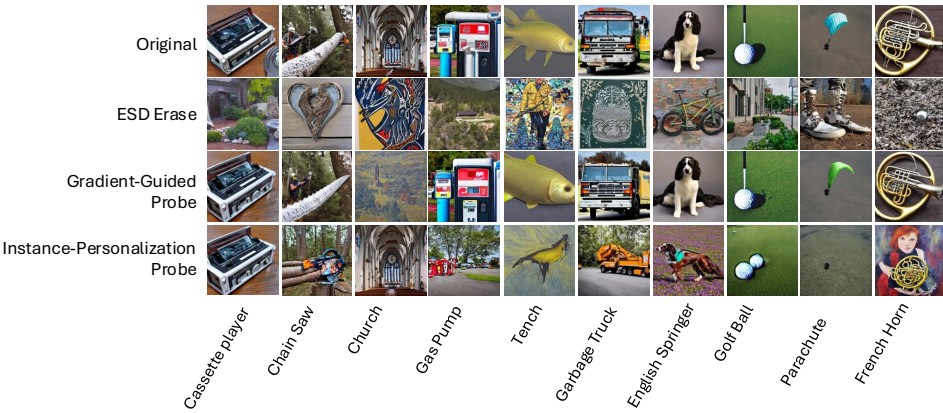

(a) ESD erasure and subsequent reactivation. The Gradient-Guided and Instance-Personalization Probes both restore erased concepts such as chain saw and french horn with high fidelity, revealing that suppression is not permanent.

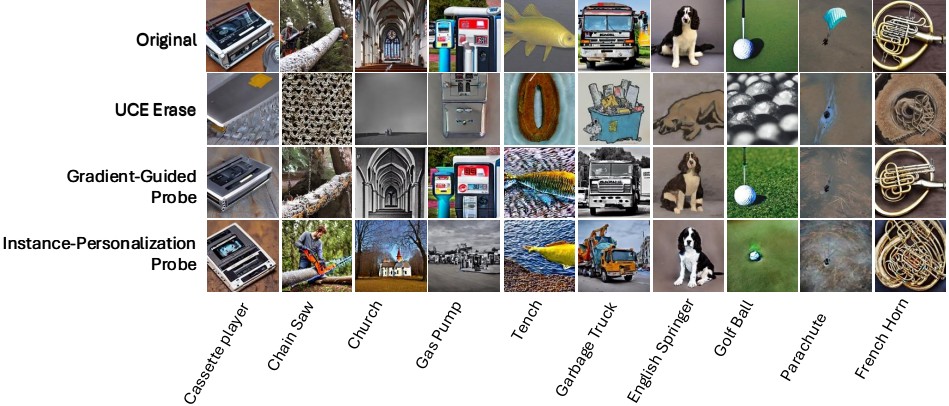

(b) UCE erasure and subsequent reactivation. Despite strong suppression under UCE, both probes successfully reinstate the erased categories, again indicating that latent representations persist in the parameter space.

Figure 1: Visual comparison of object-level erasure and reactivation under two representative methods. (a) ESD Erase and (b) UCE Erase both strongly suppress targeted concepts, visibly degrading the corresponding generations. However, our parameter-level probes (Gradient-Guided and Instance-Personalization) are able to reinstate the erased objects with high fidelity. This highlights that current erasure methods achieve conditional suppression rather than permanent removal.

## F.1 VISUALIZATION ON OBJECT CONCEPT REACTIVATION

Figure 1 provides qualitative evidence that erased object concepts remain recoverable. Under both ESD and UCE, the erased models produce generations where the targeted categories are substantially suppressed or replaced by irrelevant content. When applying our probes, the erased concepts reappear in most cases. Both the Gradient-Guided and Instance-Personalization Probes succeed in reinstating the target objects, though with slightly different visual characteristics. These consistent recoveries across multiple object categories and two distinct erasure methods reinforce our theoretical findings that residual representations persist in the parameter space and can be reinstated with minimal adaptation.

## F.2 VISUALIZATION ON ARTIST CONCEPT REACTIVATION

The visualizations in Figure 2 illustrate the effect of concept erasure and reactivation for artistic styles. Both ESD and UCE substantially suppress style-specific features, producing images that lack the distinctive attributes of Picasso, Van Gogh, Rembrandt, Warhol, and Caravaggio. However, applying either the Gradient-Guided Probe or the Instance-Personalization Probe restores the erased styles. These outcomes confirm that erasure does not fully eliminate style representations; instead, latent stylistic structures remain accessible, making reactivation feasible.

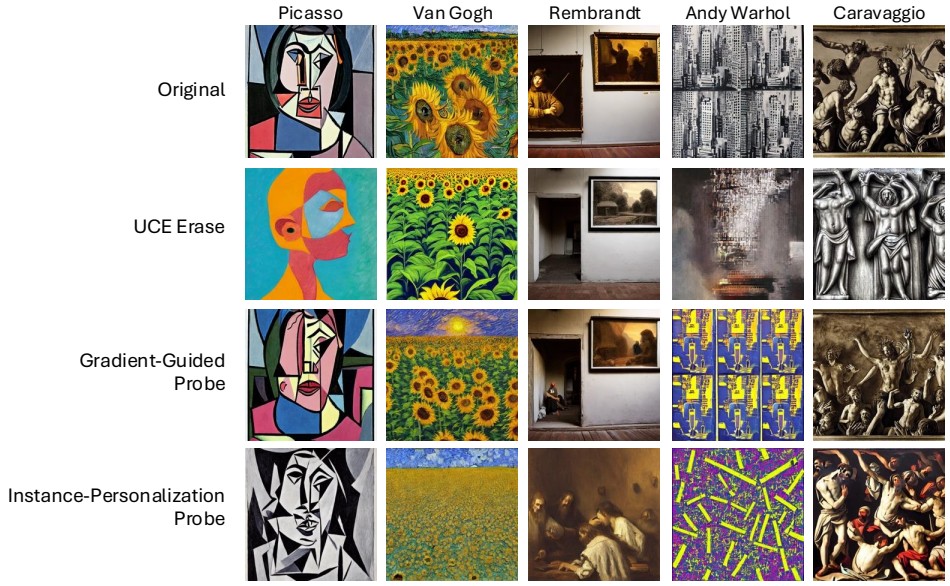

(a) Visualization of style erasure and reactivation under the ESD framework. Original generations capture the distinct styles of Picasso, Van Gogh, Rembrandt, Warhol, and Caravaggio. ESD erasure removes much of the stylistic information, while both probes recover recognizable stylistic features, though with variations in detail.

(b) Visualization of style erasure and reactivation under the UCE framework. UCE erasure substantially suppresses style-specific attributes, producing neutral outputs. Both Gradient-Guided and Instance-Personalization probes reinstate stylistic elements, demonstrating that residual representations persist despite erasure.

Figure 2: Visualization of artistic-style erasure and reactivation across five artists (Picasso, Van Gogh, Rembrandt, Andy Warhol, Caravaggio). (a) ESD-based erasure and subsequent recovery. (b) UCE-based erasure and recovery. Both Gradient-Guided and Instance-Personalization probes successfully reinstate the erased styles, though with variations in detail, illustrating that residual stylistic representations persist despite erasure.

## G  PROMPT-LEVEL AND PARAMETER-LEVEL PERSPECTIVES ON CONCEPT ERASURE

Pham et al. Pham et al. (2024) investigate concept reactivation from the prompt perspective, showing that erased concepts can be partially recovered through adversarial optimization and prompt pertur-

Table 7: Comparison of object-concept performance drop (%) between Pham (Pham et al., 2024) and our probes across three erasure methods. Numbers in parentheses indicate percentage drop relative to the original score.

| Method | Original | ESD | UCE | FMN |
|---|---|---|---|---|
| Pham et al. | 77.9 | 60.1 (**-22.8%**) | 59.4 (**-23.7%**) | 44.6 (**-42.7%**) |
| Ours | 88.85 | 81.85 (**-7.9%**) | 69.35 (**-22.0%**) | 81.65 (**-8.1%**) |

bations. Our work instead examines reversibility at the parameter level, providing a complementary view. Both studies evaluate the ESD framework on ten ImageNet categories. Pham et al. (Pham et al., 2024) report strong suppression with an average erased accuracy of 0.2% and partial reactivation at 60.1% average accuracy. As shown in Table 4 (b), our parameter-level probes achieve higher recovery, with an average reactivation accuracy of 76.7%, close to the original 74.9%. For instance, we restore "french horn" from 99.75% (original) to 100.00% (after reactivation) and "chain saw" from 77.25% to 78.25%, demonstrating that our parameter-level probes are able to reinstate erased concepts.

Table 7 provides a broader comparison across three erasure methods (ESD, UCE, FMN) for object concepts. We observe that our parameter-level approach yields smaller performance drops relative to the original model (e.g., $-7.9\%$ for ESD vs. $-22.8\%$ reported by Pham et al. (Pham et al., 2024)), and exhibits similarly lower degradation for UCE and FMN. These results indicate that parameter-level probes achieve consistently high reactivation accuracy across diverse erasure methods and reveal the residual capacity of erased models, providing a useful diagnostic tool for evaluating the robustness of concept removal.

Taken together, these comparisons suggest that while prompt-level and parameter-level analyses are complementary, parameter-level probing generally provides a relatively stronger signal of the extent to which erased representations remain recoverable, offering a more nuanced understanding of the limitations of current erasure techniques.

## H    RECOVERY WITHOUT THE PRE-ERASED MODEL

We next examine whether concept reactivation requires access to the exact pre-erased checkpoint. Our experiments show two successful pathways: using an alternative pretrained model as a guiding reference, or relying only on a small set of images that depict the target concept.

### H.1    GRADIENT-GUIDED PROBE WITH GUIDING MODELS

Recovery under the Gradient-Guided Probe does not require access to the original Stable Diffusion checkpoint. Any pretrained diffusion model that still retains the ability to generate the target concept $c$ can serve as the guiding model $\theta$. To illustrate this flexibility, we erase the concept of "church" from Stable Diffusion 1.4 and perform recovery with external guiding models, including Stable Diffusion 1.5 and DreamShaper[1]. As shown in Figure 3, the erased model $\theta''$ successfully regains the concept of "church" even when the guiding model differs from the original. The recovered outputs share high-level semantic consistency across guiding models, though subtle differences are observable, such as the shape of the tympanum above the door (blue box) and the presence or absence of fences on the grass (red boxes). This indicates that external pretrained models can serve as valid guiding references, while the actual reactivation occurs in the erased model itself. Unlike distillation, which transfers knowledge from a teacher to a student, our recovery succeeds because residual representations persist in the erased model; external models provide only auxiliary guidance.

### H.2    INSTANCE-PERSONALIZATION PROBE WITH REFERENCE IMAGES

Unlike the Gradient-Guided Probe, the Instance-Personalization Probe does not require access to the pre-erased model or any external pretrained model capable of generating the target concept $c$. Instead, it only requires a small reference set of images that visually depict the concept. These images provide direct supervision for a lightweight personalization step, allowing the erased model $\theta'$ to reacquire the erased concept from visual data. This makes the Instance-Personalization Probe applicable when pretrained models with the desired capability are unavailable, demonstrating that reactivation can be achieved from residual traces combined with limited external examples.

---

[1]https://huggingface.co/Lykon/DreamShaper

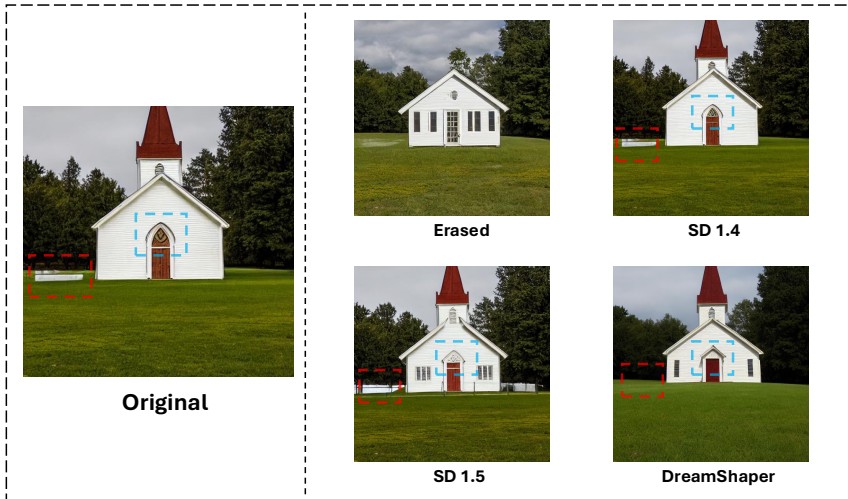

Figure 3: **Choice of guiding model** $\theta$ **for recovery.** The concept of "church" is erased from SD1.4, where the original structure is replaced with a house-like object. Recovery can be guided not only by the pre-erased model (SD1.4) but also by other pretrained models that still retain the target concept, such as SD1.5 and DreamShaper. In all cases, the erased model successfully regains the concept of "church" under the Gradient-Guided Probe, while subtle architectural differences (e.g., the tympanum above the door and fences in the foreground) vary across guiding models.

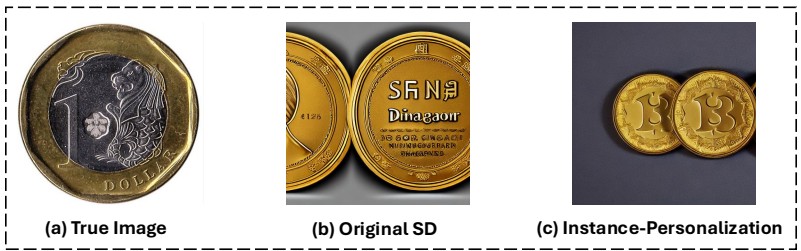

Figure 4: **A counterexample for Instance-Personalization.** The original SD model does not possess the ability to generate the coin concept, as shown in the sub-figure (b). In this case, the Instance-Personalization Probe fails to produce images resembling the ground truth in the sub-figure (c), even after fine-tuning on a few example images.

**Summary.** These two scenarios highlight complementary pathways for reactivation. The Gradient-Guided Probe leverages an external model as a guiding reference, whereas the Instance-Personalization Probe relies solely on a small reference set of images. Both confirm that the persistence of latent representations enables recovery without requiring access to the original pre-erased checkpoint.

## I WHEN CONCEPTS ARE TRULY ABSENT

An ideal erasure method would make a model behave as if it had never acquired the target concept. In such a case, few-shot personalization techniques should not be able to easily reintroduce the concept. To illustrate this, we consider a counter-example involving a specific coin image. As shown in Figure 4, the original Stable Diffusion model fails to generate convincing images of this coin, indicating that it does not contain a usable representation of the concept. Even after fine-tuning with a small set of ground-truth images using our Instance-Personalization Probe, the erased model fails to recover the target, suggesting that the concept cannot be injected without a pre-existing representational basis.

This example highlights the gap between current erasure methods and the ideal case. In practice, erased concepts are often reinstated within a few fine-tuning steps, demonstrating the persistence of residual representations. In contrast, when a concept is never present or has been fully removed, the Instance-Personalization Probe struggles to inject it effectively, and the resulting generations lack

both fidelity and generalization. This observation underscores that truly complete erasure should render reintroduction no easier than teaching the model a completely novel concept from scratch.

## J  RELATION TO UNLEARNING AND DISTILLATION

Our study is related to, but distinct from, prior research in adjacent areas. Below we clarify the differences with *machine unlearning*, its multimodal extension to *MLLM unlearning*, and *knowledge distillation*, in order to situate our contribution more precisely.

**Machine unlearning.**  Traditional machine unlearning (Xu et al., 2023; Wang et al., 2024) seeks to eliminate the influence of specific training samples so that a model behaves as if those data were never used, often driven by privacy concerns such as the "right to be forgotten." The challenge is to achieve this effect without retraining from scratch while preserving accuracy on non-erased data. Our work differs in scope: we do not address individual samples but instead test whether *concept-level representations*, such as object categories or artistic styles, remain recoverable after targeted erasure. Thus, while unlearning removes the effect of training data, we probe the persistence of semantic concepts in diffusion models.

**MLLM unlearning.**  Recent efforts extend unlearning to multimodal large language models (MLLMs) (Li et al., 2024), aiming to delete sensitive multimodal training pairs (e.g., image–text alignments) for privacy protection. This task is more complex because signals are distributed across modalities and alignment modules. Our study addresses a different dimension: rather than privacy-driven data removal, we ask whether *high-level semantic concepts* in diffusion models can be reinstated after erasure, even when they appear suppressed at the prompt level. This highlights recoverability as a limitation distinct from privacy concerns.

**Knowledge Distillation.**  While our probes involve parameter updates, they are not designed as knowledge transfer procedures. Knowledge distillation typically transfers information from a teacher model to a student model to improve accuracy or efficiency. In contrast, our probes operate entirely within the erased model and serve as controlled interventions to test whether residual representations remain activatable. Their purpose is diagnostic rather than pedagogical: they do not import new knowledge but reveal whether the erased concept still resides in the parameter space.

**Summary.**  In short, while unlearning (including MLLM unlearning) focuses on *removing the effect of sensitive training data*, and distillation focuses on *transferring knowledge*, our work investigates whether erased concepts in diffusion models are still *recoverable*. This highlights recoverability as a key consideration for assessing the robustness of erasure methods and motivates the diagnostic probes we propose.

## K  USE OF LLMS FOR WRITING ASSISTANCE

In preparing this paper, we used a large language model (ChatGPT, GPT-5, by OpenAI) to aid in the *polishing* of the manuscript text. Specifically, LLM assistance was used to:

- Improve clarity and conciseness of the draft.
- Rephrase sentences for grammatical correctness and readability.

No LLM-generated content was used without human review: all outputs were carefully checked, edited, and verified by the authors for technical correctness and consistency. No LLM was used for data generation, model training, experiment design, or result fabrication.

