# OpenReview forum: "ERASED OR DORMANT? RETHINKING CONCEPT ERASURE THROUGH REVERSIBILITY"
_ICLR.cc/2026/Conference — Submitted to ICLR 2026_

### Official Review · Reviewer_4YDp · 2025-10-31

**Soundness:** 4
**Presentation:** 4
**Contribution:** 3
**Rating:** 8
**Confidence:** 4

**Summary:**

In this paper, the authors introduces two types of diagnostic probes to test the robustness and reversibility of machine unlearning methods. One probe (Gradient-Guided Probe) intends to restore the erased concepts by reversing the suppression gradient signals of unlearned models. The other probe (Instance-Personalization Probe) aims to reinstates the erased concepts by re-binding them to a new token. Both probes provide measures for the reversibility of the concept erasure methods, and are supported by theoretical analyses.

**Strengths:**

1. The two probes proposed by the paper are intuitive and quite effective. It shed light on the current incompleteness and limitations of the existing concept erasure methods.
2. The theoretical analyses are solid. They can be used to explain the vulnerability of the existing concept erasure methods, which is rooted in the small learning rates and limited tuning steps. Furthermore, they suggest a key requirement for an irreversible erasure method is to increase the deviation effectively. These implications can guide the development of more robust concept erasure methods in the future.

**Weaknesses:**

1. The paper aims to test the robustness of SOTA concept erasure methods, but does not provide results on unlearned models that are specifically designed to be resilient to attacks, e.g., AdvUnlearn [1] by Zhang et al. It would be valuable to see how the existing defensive unlearning methods perform under two proposed probes and whether they are on the right direction towards irreversible erasure.
2. While the paper covers several concept erasure methods and related work, it does not mention other highly-relevant works, such as Chen et al. [2] and Zhang et al. [3]. [2] presents a score-distillation-based concept erasure method for one-step diffusion models while [3] uses saliency maps to efficiently finetune pretrained diffusion to forget certain concepts without compromising untargeted ones. Under the setting of unlearned one-step diffusion models, the first probe defined in Equation (1) and (2) seems not well suited.

[1] Zhang, Yimeng, et al. "Defensive unlearning with adversarial training for robust concept erasure in diffusion models." *Advances in neural information processing systems* 37 (2024): 36748-36776.

[2] Chen, Tianqi, Shujian Zhang, and Mingyuan Zhou. "Score Forgetting Distillation: A Swift, Data-Free Method for Machine Unlearning in Diffusion Models." The Thirteenth International Conference on Learning Representations.

[3] Fan, Chongyu, et al. "SalUn: Empowering Machine Unlearning via Gradient-based Weight Saliency in Both Image Classification and Generation." The Twelfth International Conference on Learning Representations. 2023.

**Questions:**

1. In Table 4 (b), why "cassette player" has a significantly worse reactivation accuracy of 9.75% than others? The original accuracy is listed as 67.50%, so this is a major drop and differs a lot from the other objects in (b) and from all the objects in (a). Could the authors explain this outlier, as it seems to contradict the claim of consistent, high-fidelity recovery?
2. How is the “Params Updated” percentage in Table 5 calculated? The implementation details state that the 'UNet attention modules' are fine-tuned, which implies a fixed set of parameters. However, the table shows this percentage increasing with iterations (e.g., 0.30% to 1.38% for “chain saw”). Why does this percentage increase, and is some unstated sparse update being applied?
3. Is it possible to generalize the probes for unconditional diffusion models?

---

### Official Review · Reviewer_oZ8L · 2025-11-02

**Soundness:** 2
**Presentation:** 2
**Contribution:** 2
**Rating:** 4
**Confidence:** 5

**Summary:**

The paper proposes two gradient-based recovery approaches to restore unlearned concepts removed by concept unlearning methods: a Gradient-Guided Probe, which restores suppressed behavior by reversing the gradient signal, and an Instance-Personalization Probe, which reinstates concepts through personalization. The paper shows that existing unlearning methods do not fully eliminate concepts but merely push them below the surface, allowing them to be recovered.

**Strengths:**

-	The paper presents an interesting theoretical result: the expected squared difference between the unlearned and recovered models converges to a steady-state bound, implying that the difference between their parameters remains bounded

**Weaknesses:**

-	Writing issue: The statement in line 151 — “as a result, erased concepts are often conditionally suppressed rather than fully removed” — is not well supported by experimental or theoretical evidence.
-	The Instance-Personalization Probe is conceptually similar to the approach in [1] (which uses DreamBooth rather than Textual Inversion, as in [1]). It inherits a critical limitation: the strong dependence of personalization quality on the choice of the reference set $\mathcal{X}_{ref}$.
-	Using reference data from the erased (unwanted) concepts to recover those same concepts seems unfair and somewhat trivial. Given the power of personalization methods, a model can relearn a concept from only a few samples. Thus, it becomes unclear whether the probe recovers an unlearned concept or simply injects a new one into the model?
-	The gradient-guide probe update in Eq (1) requires both erased model $\theta’$ and the original model $\theta^{‘’}$. However, if the original model is still available and capable of generating the undesired concepts, the motivation for this recovery approach becomes less compelling

[1] Pham, Minh, et al. "Circumventing concept erasure methods for text-to-image generative models." arXiv preprint arXiv:2308.01508 (2023).

**Questions:**

In addition to the concerns raised above, could the authors help to address the following points:

-	In Equation (2), what are the $\theta^{‘}$ and $\theta^{‘’}$.
-	If the gradient-based probe does not require the original model, how can we distinguish between recovering an unlearned concept and learning a completely new concept unknown to the model (this differ from Instance-Personalization Probe where reference data is available). For example, can we use gradient-based probe to learn “astronaut monkey” concept or something that totally new to the model?
-	How does this work compare to the recent paper [1], which also demonstrates that fine-tuning on unrelated tasks can inadvertently recover unlearned concepts?
-	How does this paper relate to [2], which conceptually distinguishes between two forms of unlearning: guidance confusion and density destruction?
-	What is the impact of the proposed recovery methods on unrelated concepts—does the recovery process unintentionally alter them?

[1] George, Naveen, et al. "The Illusion of Unlearning: The Unstable Nature of Machine Unlearning in Text-to-Image Diffusion Models." CVPR 2025

[2] Lu, Kevin, et al. "When Are Concepts Erased From Diffusion Models?." NeurIPS 2025

---

### Official Review · Reviewer_QHjo · 2025-11-03

**Soundness:** 2
**Presentation:** 3
**Contribution:** 2
**Rating:** 2
**Confidence:** 4

**Summary:**

This paper investigates whether concept erasure methods in diffusion models truly remove targeted concepts or merely suppress them. The authors introduce a diagnostic framework with two parameter-level probes: a Gradient-Guided Probe that reverses suppression gradients and an Instance-Personalization Probe that uses few-shot learning. Evaluations on on six erasure methods reveal that erased concepts can be consistently recovered with minimal parameter updates. Theoretical analysis provides bounds showing reactivated models remain close to original unerased models.

**Strengths:**

* The paper is generally well written and the problem of whether current erasure methods are doing suppression is well-motivated.
* The theoretical analysis showing that the reactivated model remain close to the unerased model is appreciated.

**Weaknesses:**

* Since all of the probing techniques all update the weights, I do not see how this is different from simply fine-tuning the model to learn the supposedly erased concepts.

* It is unclear to me whether the reactivation of the erased concepts is because the concepts are not fully erased, or simply because the probing techniques are very good at making the model learn concepts.

**Questions:**

* How do the probing methods compare to standard fine-tuning using the same number of parameters?
* Can you show that the reactivation of the erased concepts are truly due to the ineffectiveness of erasure methods?For instance, can you show either on a toy model or on SD 1.4 that it’s harder to probe a concept that the model truly does not know about, compared to a concept that is erased?

---

### Official Review · Reviewer_Ruij · 2025-11-06

**Soundness:** 3
**Presentation:** 2
**Contribution:** 2
**Rating:** 4
**Confidence:** 4

**Summary:**

This paper studies the revertability of erasing concepts in model unlearning for diffusion models. Specifically, it proposes two lightweight probe methods that rebind erasing concepts from a few example.

**Strengths:**

- The proposed probe methods to erasing concept revertability make sense.
- The experiments are quite intensive with 6 popular erasing methods for diffusion models.

**Weaknesses:**

- This paper seems to reinvent the wheel because it is well-known that erasing concepts can be easily restored using personalized AI methods such as Textual Inversion or DreamBooth. Moreover, some recent papers indicated that quantization can restore erasing concepts or even fine-tuning on different classes/concepts can restore erasing class/concept.
-  Moreover, theoretical analysis of reactivation bound is not clearly presented. Although the results in Theorems 2 and 3 make sense and quite trivial, it is still unclear how the SDEs in Line 225 and 227 are relevant to  the update in Gradient-Guided Probe.
-  It is unclear about the main claims in this paper
   - If its claim is about the easy revertability of erasing concepts, it reinvents the wheel.
   - If its claim is about new approaches to recover erasing concepts from unlearned models, it should conduct experiments to compare to existing methods for recovering erasing concepts.

**Questions:**

- Eq. (1) is relevant to ESD. Please explain why it helps recover erasing concepts.
- In Instance-Personalization Probe, similar to DreamBooth, you fine-tune entire models? Moreover, do you learn rare token $v_*$?

---

### Meta-Review · Area_Chair_XFqf · 2026-01-11

**Summary:**

This paper studies the revertability of erasing concepts in model unlearning for diffusion models. Specifically, it proposes two lightweight probe methods that rebind erasing concepts from a few example.

**Reviewer Concerns:**

Authors did not submit the rebuttal.

Reviewers main concerns are related to limited novelty and unsubstantiated claims.

- This paper seems to reinvent the wheel because it is well-known that erasing concepts can be easily restored using personalized AI methods such as Textual Inversion or DreamBooth. Moreover, some recent papers indicated that quantization can restore erasing concepts or even fine-tuning on different classes/concepts can restore erasing class/concept.

- It is unclear to me whether the reactivation of the erased concepts is because the concepts are not fully erased, or simply because the probing techniques are very good at making the model learn concepts.

- The paper does not discuss related work on robust unlearning methods. It would be valuable to see how the existing defensive unlearning methods perform under two proposed probes and whether they are on the right direction towards irreversible erasure.

**Reviewer Scores:**

All reviewers gave scores of 4
Authors did not submit a rebuttal.

---

### Decision · Program_Chairs · 2026-01-26

Reject